# Automotive Lidar Modelling Approach Based on Material Properties and Lidar Capabilities

**DOI:** 10.3390/s20113309

**Published:** 2020-06-10

**Authors:** Stefan Muckenhuber, Hannes Holzer, Zrinka Bockaj

**Affiliations:** 1Virtual Vehicle Research GmbH, Inffeldgasse 21A, 8010 Graz, Austria; hannes.holzer@v2c2.at; 2Infineon Technologies Austria AG, Babenbergerstrasse 10, 8020 Graz, Austria; zbockaj@hotmail.com

**Keywords:** lidar, sensor model, material reflectance, virtual testing, automotive

## Abstract

Development and validation of reliable environment perception systems for automated driving functions requires the extension of conventional physical test drives with simulations in virtual test environments. In such a virtual test environment, a perception sensor is replaced by a sensor model. A major challenge for state-of-the-art sensor models is to represent the large variety of material properties of the surrounding objects in a realistic manner. Since lidar sensors are considered to play an essential role for upcoming automated vehicles, this paper presents a new lidar modelling approach that takes material properties and corresponding lidar capabilities into account. The considered material property is the incidence angle dependent reflectance of the illuminated material in the infrared spectrum and the considered lidar property its capability to detect a material with a certain reflectance up to a certain range. A new material classification for lidar modelling in the automotive context is suggested, distinguishing between 7 material classes and 23 subclasses. To measure angle dependent reflectance in the infrared spectrum, a new measurement device based on a time of flight camera is introduced and calibrated using Lambertian targets with defined reflectance values at 10%, 50%, and 95%. Reflectance measurements of 9 material subclasses are presented and 488 spectra from the NASA ECOSTRESS library are considered to evaluate the new measurement device. The parametrisation of the lidar capabilities is illustrated by presenting a lidar measurement campaign with a new Infineon lidar prototype and relevant data from 12 common lidar types.

## 1. Introduction

Advanced driver assistance system and automated driving (ADAS/AD) functions will provide many benefits such as improved passenger safety and comfort, reduced congestion and emissions and so forth. ADAS/AD functions will furthermore enable new transportation use cases and applications [1]. To derive a suitable driving behaviour and respond to changing surrounding conditions, ADAS/AD functions must rely on environment perception sensors such as lidar (light detection and ranging), radar and camera. In particular, automated driving functions have a high demand regarding environment perception, that is, the sensor system need to provide a precise 3D map of the vehicle’s surrounding. Diverse and redundant sensor types are required to enable a robust environment perception during all possible weather conditions. A combination of lidar, radar and camera is considered to eventually provide the necessary capabilities to fulfil these high demands. Lidar sensors play an essential role in this sensor suite, since lidars provide a depth map with improved angular resolution compared to radar [2,3]. High costs of mechanically spinning lidars are currently a limiting factor, however prices are going down with new technologies like MEMS based mirrors, optical phased array and so forth [2,4]. For example, Druml et al. [5] presented a lidar prototype that shall enable commercial lidar sensors with a range of more than 200 m for costs less than $200.

Development and validation of reliable environment perception systems suitable for ADAS/AD functions represent major challenges in todays automotive industry. Increasing requirements with respect to costs, safety, and development time demand the extension of conventional test methods, for example, physical test drives, with simulations in virtual test environments [1,6]. The environment perception system is simulated in a virtual test environment by sensor models. Hence, sensor models are an important prerequisite for virtual testing of vehicles with ADAS/AD functions.

The flow chart in Figure 1 illustrates the data flow of object based and raw data based lidar models in a virtual test environment for ADAS/AD functions. An environment simulation, like Vires VTD [7], IPG CarMaker [8], CARLA [9] or AirSim [10] simulates the test scenario including infrastructure, traffic participants, environment conditions and so forth. The true state of the environment, called ground-truth, is forwarded to the sensor model either as object list or as complete 3D scenario. An object based lidar model modifies the ground-truth object list according the sensing capabilities of the specific lidar, for example, field of view. A raw data based lidar model creates a point cloud based on the 3D scenario. The output of the lidar model serves as input for the ADAS/AD function under test. Up to now, there is no standard format for the interfaces between virtual environment, sensor model and ADAS/AD function. A promising approach by Hanke et al. [11], called Open Simulation Interface (OSI), is currently under development, however the multitude of both commercial and open-source virtual environments and sensor specific object classification procedures propose considerable challenges.

Previous work on object-based sensor models include Hanke et al. [12], Stolz and Nestlinger [13] and Muckenhuber et al. [14]. Hanke et al. [12] suggest a modular architecture for object based sensor models. The ground-truth object list from the virtual environment is modified sequentially by a number of modules. Each module represents a specific sensor characteristic or environmental condition. Stolz and Nestlinger [13] present a generic sensor model to simulate a perfect or zero-error sensor on object level. The sensor models recognises all objects in the sensor’s field of view correctly. Object parameters that influence the perception process and disturbing environmental conditions, like unfavourable weather conditions, are not considered. Muckenhuber et al. [14] introduce an object based sensor model including coverage based on a simple ray tracing approach, object dependent field of views and false negative/false positive detections based on probabilistic relationships.

Complex lidar models such as DIRSIG provide very detailed lidar raw data including atmospheric and thermodynamic components (http://www.dirsig.org), but are computationally demanding and not implemented into virtual test environments for testing ADAS/AD functions. Automotive raw data based lidar models, for example, Hanke et al. [15], typically generate point clouds based on computationally efficient ray-tracing and rendering methods. Several environment simulations are already capable of providing a perfect lidar point cloud without considering object parameters and disturbing environmental conditions and a few environment simulations (e.g., https://www.tesis.de/en/sensorsimulation/) are advertising reflection intensity simulation based on angle between laser beam and object surface and material properties. However, to our knowledge little is documented on how and which material parameters are used for deriving these intensity values. In conclusion, current automotive lidar models typically provide a reduced ground-truth object list or perfect point cloud without detailed information on considered material properties and corresponding lidar capabilities. We propose a novel lidar modelling approach that includes angle dependent material reflectance and corresponding lidar capabilities based on spectral reflectance $R_{\lambda }$ relative to Lambertian targets (Figure 1). Other effects that have an impact on the range performance of a lidar, exceed the scope of this publication. For example, atmospheric effects, such as rain and fog [16,17], and other environmental effects, such as solar radiation received by the lidar or modification of the backscattering characteristics of a wet surface, can considerably influence the lidar range performance. To eventually enable virtual homologation of ADAS/AD functions, these effects need to be considered and modelled in addition. The inclusion of environmental effects into the presented lidar modelling approach will be discussed in perspectives in Section 9.

Angle dependent material reflectance as considered in the presented lidar modelling approach have been subject to previous studies. For example, Reference [18] measured angle dependent backscattering properties and hemispherical reflectance of maple wood, red brick, concrete brick, and asphalt in a laboratory setup in the visible (488 nm), near-infrared (1320 nm), and mid-infrared (7600 nm) spectral ranges. To be able to collect relevant reflectance measurements of materials and objects that cannot be easily brought into a laboratory, we introduce a new portable measurement device for collecting angle dependent reflectance values $R_{\lambda }$ in % relative to Lambertian targets at a wavelength of λ = 945 nm, which is very close to the wavelength of most lidar types.

Section 2 and Section 3 introduce the considered material properties and lidar capabilities and suggest corresponding measurement concepts. Section 4 introduces the lidar modelling approach and explains how material properties and lidar capabilities can be included into both object and raw data based lidar models. Section 5 suggests a material classification for lidar modelling in the automotive context. Section 6 introduces a new measurement device based on a time of flight camera to measure angle dependent reflectance in the infrared spectrum. Section 7 evaluates the measurements of the new device using reflectance data from the NASA ECOSTRESS library. Section 8 elaborates on the parametrisation of the lidar capabilities in the modelling approach by presenting a lidar measurement campaign and relevant data from common lidar types. Section 9 completes the paper with a conclusion and gives an outlook on future work.

## 2. Material Properties

The radiometric response of a material can be described by the bidirectional reflectance distribution function (BRDF) expressed as $f_{r}\left(\omega_{r},\omega_{i}\right)$ with $\omega_{i}$ being the direction of the incoming radiation, and $\omega_{r}$ being the reflection direction [19]. The unit of the BRDF is 1/sr and it is defined as the relationship between the reflected spectral radiance $L_{r}\left(\omega_{r}\right)$ reflected by the material into the direction of the sensor and the incidence spectral irradiance $E_{i}\left(\omega_{i}\right)$ received by the surface:(1)$$f_{r}\left(\omega_{r},\omega_{i}\right)=\frac{dL_{r}\left(\omega_{r}\right)}{dE_{i}\left(\omega_{i}\right)}=\frac{dL_{r}\left(\omega_{r}\right)}{L_{i}\left(\omega_{i}\right)cos\left(\theta_{i}\right)d\omega_{i}}\left[\frac{1}{sr}\right].$$

The spectral radiance in wavelength $L_{x}\left(\omega_{x}\right)$ is defined as the radiant flux emitted, reflected, transmitted or received by a surface per unit solid angle, area and wavelength:(2)$$L_{x}\left(\omega_{x}\right)=\frac{\partial^{3}\Phi }{\partial \lambda \partial \omega_{x}\partial Acos\left(\theta \right)}\left[\frac{W}{m^{3}sr}\right].$$

The corresponding SI unit of $L_{r}$ is watt per steradian per metre exponent three. Φ is the radiant flux in watts, λ the wavelength in meters, $\omega_{x}$ the solid angle in steradian, Acos(θ) the projected area in square meters and θ the incidence angle.

The incidence spectral irradiance $E_{i}$ is defined as is the radiant flux received by a surface per unit area and wavelength and the corresponding SI unit of $E_{i}$ is watt per metre exponent three.

Considering a monostatic lidar, laser emitter and receiver are mounted very close to each other in a single unit. This means, if other external sources of irradiance are neglected, that the direction $\omega_{i}$ of the incidence spectral irradiance $E_{i}$ and the direction $\omega_{r}$ of the reflected spectral radiance $L_{r}$ are always equal. Assuming that the BRDF of the considered material remains constant if the material is rotated parallel to its surface allows to further simplify the BRDF. In this case, the radiometric response of a specific material depends mainly on the used wavelength λ and the incidence angle θ. The used wavelength λ is defined by the lidar hardware setup and typically constant. This allows to express the radiometric response of a material by a function $f_{\lambda }\left(\theta \right)$. In the following, the expression spectral reflectance $R_{\lambda }\left(\theta \right)$ is used to describe the radiometric response of a specific material:(3)$$R_{\lambda }=f\left(\theta \right)\left[\%\right].$$

In the following, $R_{\lambda }$ will be given in % compared to a 100% Lambertian target at $\theta =0^{\circ }$ (NB: a Lambertian target is an ideal diffusely reflecting surface, quasi-invariant spectrally with a backscatter coefficient R/π with *R* being its spectral hemispheric reflectance). For example, a value of $R_{\lambda =905\mathrm{nm}}\left(\theta =45^{\circ }\right)=50\%$ means that, at a wavelength λ = 905 nm and at an incidence angle $\theta =45^{\circ }$, the respective material has the same reflectance behaviour as a Lambertian target with $R_{\lambda =905\mathrm{nm}}\left(\theta =0^{\circ }\right)=50\%$ at an incidence angle $\theta =0^{\circ }$. Note that values above 100% are possible, since some materials (e.g., retroreflectors) concentrate the reflected radiance back towards the illumination source.

To describe the radiometric response of a material by using spectral reflectance $R_{\lambda }$ given in % relative to Lambertian targets has two major reasons:The specification sheets of common lidar types [20,21,22,23,24,25,26] describe the lidar performance depending on Lambertian target reflectance in %. Using spectral reflectance $R_{\lambda }$ given in % relative to Lambertian targets allows to directly link the lidar performance given in the specification sheet with the presented material measurements.Well defined Lambertian targets are commercially available [27] and compared to specular targets less prone to angle inaccuracies during measurement campaigns. This makes Lambertian targets a good choice both for calibration of the newly developed measurement device, introduced in Section 6, and for lidar parametrisation, as further illustrated in Section 8.

### Measurement Concept

To measure the angle dependent reflectance $R_{\lambda }$ of a certain material, a measurement device with a laser transmitter and receiver operating at the wavelength λ are pointed towards the considered material and moved at a certain distance from $\theta =0^{\circ }$ to $\theta =90^{\circ }$. The laser receiver must provide a value expressing the amount of backscattered light, such as the intensity *I*, to eventually derive the reflectance $R_{\lambda }$ of the illuminated surface. The measurement device is calibrated using Lambertian targets. The spectral reflectivity $R_{\lambda }$ of a certain material can then be measured according to the following expression: The intensity signal $I_{\lambda ,\theta }$ measured at the wavelength λ, at an incidence angle θ on the material is divided by the intensity signal $I_{L,\lambda ,\theta =0^{\circ }}$ measured by the same measurement device on a Lambertian target in nadir incidence ($\theta =0^{\circ }$) multiplied by the hemispherical spectral reflectance value $R_{L,\lambda }$ of the Lambertian target at wavelength λ:(4)$$R_{\lambda }\left(\theta \right)=\frac{I_{\lambda ,\theta }}{I_{L,\lambda ,\theta =0^{\circ }}}*R_{L,\lambda }\;\left(\%\right).$$

Figure 2 illustrates the proposed measurement concept to derive the angle dependent reflectance $R_{\lambda }$ of a material. Figure 3 depicts a corresponding exemplary reflectance $R_{\lambda }$ curve depending on the incidence angle θ.

## 3. Lidar Capabilities

The considered lidar property is the lidar’s capability to detect a material with a certain reflectance value $R_{\lambda }$ up to a certain maximum detection range $d_{x}$. The wavelength λ is here the wavelength of the considered lidar. The minimum reflectance value that can still be detected at a certain range *r* is then called reflectance limit $R_{L}$. The reflectance limits $R_{L}$ of a lidar typically increase with range *r*.

There exits several different definitions for the maximum detection range $d_{x}$ depending on reflectance $R_{\lambda }$ from different perspectives of optics, system engineering, as well as perception. In the following, the maximum detection range $d_{x}$ is only based on a single reflectance threshold value and can therefore be given as discrete value in *m*. Note that this simplifies the the detectability to 0 and 1, whereas in reality the detection is more close to a probability over a wide range.

A lidar can detect a certain material if the return signal exceeds the detection threshold of the lidar. The strength of the lidar return signal varies according to the material reflectance $R_{\lambda }$ divided by the range to the power of two, three or four depending on the relative size of the laser beam compared to the size of the illuminated surface (Figure 4). If the area of the laser beam is smaller than the illuminated surface in both dimensions, the lidar return signal decreases proportional to $1/r^{2}$. If the area of the laser beam is smaller than the illuminated surface in one dimension and larger in the other dimension, the lidar return signal decreases proportional to $1/r^{3}$. If the area of the laser beam is larger than the illuminated surface in both dimensions, the lidar return signal decreases proportional to $1/r^{4}$. In the following, we assume that the laser beam is always smaller than the illuminated surface and therefore a return signal attenuation proportional to $1/r^{2}$.

### Measurement Concept

Figure 5 illustrates the proposed measurement concept to derive the reflectance limits $R_{L}$ of a lidar. Targets with defined reflectance values $R_{\lambda }$ are placed at different distances to estimate the maximum detection range $d_{x}$ for each target, and hence, for the corresponding reflectance value $R_{\lambda }$. To minimise the impact of wavelength and target orientation, Lambertian targets $L_{x}$ with ideal diffusely reflecting surfaces are used oriented normal to the lidar, that is, with incidence angle $\theta =0^{\circ }$. The reflectance limits $R_{L}$ of the lidar are then equal to the reflectance values $R_{x}$ of the targets at the corresponding maximum detection range $d_{x}$.

A quadratic interpolation
(5)$$R_{L}\left(r\right)=a+b\times r^{2},$$
between the measurement points and towards 0% reflectance at 0 m range, and a linear cut-off at the measurement point with the highest reflectance value (in this example $R_{\lambda }\left(L_{N}\right)$) are applied to cover all possible reflectance values. Figure 6 illustrates an exemplary reflectance limits function $R_{L}\left(r\right)$ depending on range *r* that is inter- and extrapolated to cover all reflectance values. As further elaborated in Section 4, the function $R_{L}\left(r\right)$ divides the detection area (green area in Figure 6) and the undetected area (red area in Figure 6).

## 4. Lidar Modelling Approach

The following Section describes the proposed lidar modelling approach that allows to include material properties and corresponding lidar capabilities into both object based and raw data based lidar models.

As described in Section 2, the considered material property is the angle dependent reflectance $R_{\lambda }\left(\theta \right)$. Section 3 introduced corresponding lidar capabilities, that is, the reflectance limits $R_{L}$. Figure 7 illustrates how the spectral directional reflectance $R_{\lambda }\left(\theta \right)$ of the material and the reflectance limits $R_{L}$ of the lidar can be included into a lidar model.

Each material in the environment simulation is assigned a reflectance function $R_{\lambda }\left(\theta \right)$ function that depends on the incidence angle θ. By calculating the angle between laser beam direction and illuminated surface, the reflectance value $R_{\lambda }$ of the illuminated material is derived. By calculating the distance between lidar and illuminated material, that is, the range *r*, the corresponding reflectance limit $R_{L}$ is derived from the lidar capabilities. Depending on whether the reflectance limit $R_{L}$ is above or below the reflectance $R_{\lambda }$, the illuminated material is detected or remains undetected:(6)$$\begin{array}{c} R_{L}\left(r\right)>R_{\lambda }\left(\theta \right)\to \mathrm{detected}\\ R_{L}\left(r\right)<R_{\lambda }\left(\theta \right)\to \mathrm{undetected}. \end{array}$$

In object based lidar models, each object gets assigned a reflectance function $R_{\lambda }\left(\theta \right)$ as a function of incident angle θ. In raw data based lidar models, the reflectance $R_{\lambda }\left(\theta \right)$ function is defined for each material.

## 5. Material Classification

Nature provides a wide range of different materials. However, not all materials are relevant for automotive simulations and several materials have similar reflectance properties in the near-infrared spectrum. Materials with similar properties can be grouped together into a single material class without significant impact on the simulation results. Classifying materials is also a requirement of environment simulations, since they typically build a new environment based on a certain set of objects and corresponding materials. The alternative, to build a digital twin based on real measurements of each included material surface, is very uncommon due to the related high cost and time effort.

Therefore, we suggest a material classification (Table 1), that includes the most relevant materials for lidar simulations in the automotive context, and a classification schema, that allows to choose between different levels of detail, that is, material distinction can be done by class or subclass. For each material class and subclass, mean reflectance values incl. standard deviations, minima and maxima as a function of incidence angle and wavelength need to be derived from a representative set of samples and stored in a reflectance database. To fill the reflectance database, measurements from a time-of-flight (TOF) camera mounted on an angle adjustment device (Section 6) were collected and evaluated against reflectance values from the NASA ECOSTRESS spectral library (Section 7).

By assigning a material class or subclass to each object or material surface in the environment simulation, the reflectance database as well as the presented lidar modelling approach can be easily included into the virtual test environment (see Section 4).

## 6. TOF Camera Measurements

To provide reflectance values for the material classes and subclasses listed in Table 1, reflectance measurements were collected using a TOF camera, that is capable of recording intensity values. The used TOF camera is a ’FusionSens Maxx GN8-1XNBA1 60 outdoor’, which operates at 945 nm and builds on the Infineon TOF sensor ’Infineon IRS1125C’ and a 2 W VCSEL illumination source [28]. The viewing angle of the TOF camera is $60^{\circ }$ × $45^{\circ }$, the resolution of the resulting image is 352 × 287 pixels, the measurement range is 0.2–4 m, the depth resolution is 1–2% and the frame rate is programmable between 5–60 fps.

Being an active sensor in the near infrared spectrum (945 nm), the used TOF camera is expected to experience a similar angle dependent reflectance behaviour by the illuminated material as a typical automotive lidar. Having a well defined and constant laser source (2 W VCSEL) in the TOF camera, this allows to derive the material’s reflectance. To keep the distance to the material at a constant value and observe angle dependent reflectance values, the TOF camera was mounted on an angle adjustment device, that was specifically designed and built for this purpose.

Similar to a lidar, the TOF camera works by measuring the travel-time of the modulated light from the active light source to the scene and back to the sensor and by measuring the intensity of the received light pulse. Each pixel in the resulting depth image represents the distance between the camera and the corresponding objects. Each value in the intensity image represents the reflection properties of the illuminated surface. For our use case, the main feature of the TOF camera is its capability to record intensity images.

Figure 8 illustrates the measurement principle of the TOF camera including all relevant processing units and exemplary depth and intensity images depicting a Lambertian target on a non-reflecting surface. The working principle of a TOF camera starts with an active illumination unit, that emits pulsed infrared light at 945 nm. A part of the emitted light is reflected and travels back to the camera, where it is projected by a lens on an array of photodiodes. The phase of the incoming pulsed light signal is shifted due to the travel time. The phase shift between sent and reflected light pulse is measured in each pixel by a photonic mixer device (PMD). The PMD transfers the generated charge of the photodiode either into capacitor A or B, depending on the modulation signal frequency $F_{mod}$, which also determines the frequency of the emitted pulsed light signal. A phase value $p_{i}$ is calculated by subtracting the collected voltages of A and B. Taking four images of the same scene with $0^{\circ }$, $90^{\circ }$, $180^{\circ }$, and $270^{\circ }$ phase shift of the emitted light pulse allows to derive both the distance *d* between TOF camera and object and intensity of the reflected light *I* [29]:(7)$$\phi =arctan\frac{p_{90^{\circ }}-p_{270^{\circ }}}{p_{0^{\circ }}-p_{180^{\circ }}}\;\;\;\to \;\;\;d=\frac{c}{2}*\frac{\phi }{2\pi F_{mod}},$$
(8)$$I=\frac{\sqrt{{\left(p_{180^{\circ }}-p_{0^{\circ }}\right)}^{2}+{\left(p_{90^{\circ }}-p_{270^{\circ }}\right)}^{2}}}{2}.$$

To ensure robust operation of the TOF camera both in indoor and outdoor applications and provide reliable material reflectance values also during sunlight, the influence of background light must be suppressed. Background light is added on the reflected signal as signal offset. Since the phase value $p_{i}$ is calculated by subtracting the voltages of A and B, and the offset is equally present in both measurements, the influence of background light is already minimised by the measurement principle itself. To further suppress background illumination, the TOF camera uses optical filters, burst operation mode (same average power, but higher peak power than in continuous operation) and a special circuitry which directly compensates for any active constant offset, called SBI (suppression of background illumination) [30].

Figure 9 illustrates the measurement setup including the TOF camera, the angle adjustment device for setting the incidence angle θ and a PC for data recording. The angle adjustment device allows set the TOF camera orientation to any incidence angle θ between $0^{\circ }$ and $90^{\circ }$. The complete measurement setup is portable and therefore well suited for collecting measurements on materials and objects that cannot be easily brought into the laboratory, for example, asphalt.

As stated above, the TOF camera has a viewing angle of $60^{\circ }$ × $45^{\circ }$, which represents a more divergent laser source than typical automotive lidar systems. In order to represent a lidar with less divergence, only a few center pixels of the TOF image are used for evaluation. The close distance to the target and the consideration of only a few center pixels result in a small footprint, similar to divergent laser sources as used by typical automotive lidar systems. As described above, the resulting TOF image has a resolution of 352 × 287 pixels. A line of 40 pixels (pixels 156–196 out of 352) normal to the tilting angle in the center of the image (row 145 of 287) has been used for evaluation. At each incidence angle, a total of 300 measurements were taken with a frame rate of 5 fps. The first 200 measurements were dismissed to allow both laser source and receiver to reach a good operating temperature. The intensity value *I* is then calculated as average of the 40 center pixels of the last 100 measurements, that is, intensity *I* represents the mean value of 4000 single measurement points.

To calibrate the measurement setup, Lambertian targets with defined reflectance values $R_{\lambda }\left(L\right)$ at 10%, 50%, and 95% [27] were used. To suppress potentially disturbing reflections by the surrounding material, the Lambertian targets were placed on a non-reflecting surface (Figure 9). As described above, the TOF camera measures an intensity value *I*. The resulting intensity values *I* depending on incidence angle θ and reflectance value $R_{\lambda }\left(L\right)$ are depicted in Figure 10. The intensity values *I* decrease with increasing incidence angle θ approximately according to cos(θ) (Figure 10a). This is expected, since the amount of illumination hitting the observed surface is reduced with increasing incidence angle θ by cos(θ).

In case of the 10% Lambertian target, higher values than suggested by the theoretical curve $I\left(\theta =0^{\circ }\right)\times cos\left(\theta \right)$, were measured at high incidence angles ($\theta =60^{\circ },70^{\circ },80^{\circ }$). This could have two reasons. First, the return signals of these particular measurements are relatively weak and closer to the noise floor of the TOF camera than the other measurements, which means that thermal noise and other disturbing effects might have a greater impact in this case. Second, the Lambertian targets are about 1 cm high and there might be disturbing reflections coming from the edges of the Lambertian target that could increase the intensity value *I* at high incidence angles. Measurements at high incidence angles ($\theta =60^{\circ },70^{\circ },80^{\circ }$) of targets with very low reflectance (order of 10% or less) must therefore be considered with a higher uncertainty and a potential bias of around 1–5%.

Based on the calibration measurements shown in Figure 10, the relationship between intensity value *I* and reflectance value *R* (%) of a Lambertian target at incidence angle $\theta =0^{\circ }$ can be expressed as:(9)R(%)=−0.2130+0.0698×I.

Intensity values were collected for materials listed in Table 1 at incidence angles $\theta =[0^{\circ },10^{\circ },20^{\circ },30^{\circ },40^{\circ },50^{\circ },60^{\circ },70^{\circ },80^{\circ }$] and reverted to corresponding reflectance values using Equation (Equation 9). The reflectance values are stored together with an image that depicts the observed material with the measurement setup. Examples are shown in Figure 11.

## 7. Evaluation of TOF Camera Measurements

The following Section evaluates the TOF camera measurements presented in Section 6 and addresses two major questions:To which extent can TOF camera measurements at 945 nm be used to represent the material reflection behaviour at other wavelengths that are common for lidar?How large is the spread within a subclass and how representative are single TOF camera measurements to capture the reflection behaviour of an entire material subclass?

To address these questions, data from the NASA ECOSTRESS spectral library [31] were analysed and compared to the TOF camera measurements.

The ECOSTRESS spectral library version 1.0 was established by Meerdink et al. [32] to support the ECOsystem Spaceborne Thermal Radiometer Experiment on Space Station (ECOSTRESS) mission, which was launched in June 2018 to measure plant temperatures and better understand how plants respond to stress. The ECOSTRESS library represents an expansion of the Advanced Spaceborne Thermal Emission Reflection Radiometer (ASTER) spectral library [33], and includes more than 3400 spectra of lunar and terrestrial soils, man-made materials, meteorites, minerals, vegetation, rocks, and water/snow/ice.

Based on this extensive material library, several material parameters relevant for the automotive lidar use case can be extracted. A total of 488 spectra have been considered relevant for the automotive lidar use case and could be related to the material subclasses metal, glass, rubber, asphalt, concrete, wood, rock, photosynthetic vegetation, and non-photosynthetic vegetation. The amounts of spectra relevant for each individual subclass are shown in Table 1.

The considered wavelength range (350–15, 400 nm covered with 20 nm steps) of the ECOSTRESS spectral library covers all common lidar wavelengths, which are typically between 830 nm and 940 nm, apart from a few exceptions at 660 nm and 1550 nm. Figure 12 depicts an exemplary spectrum of hemispherical reflection (for the material subclass photosynthetic vegetation) based on the ECOSTRESS library including the wavelengths of common lidar types and the used TOF camera. The same spectrum of hemispherical reflection including minimum, maximum, mean and standard deviation values was derived for all material subclasses that could be related to data from the ECOSTRESS library.

To evaluate whether the TOF camera measurements at 945 nm can be used to represent the material reflection behaviour at other wavelengths of common lidar types, the above mentioned 488 spectra of hemispherical reflectance from the ECOSTRESS library, that could be related to 9 material subclasses, were analysed. The difference between the hemispherical reflectance value at 940 nm (close to TOF camera) and the hemispherical reflectance value at 660 nm (close to Leica P30/P40/P50), 840 nm (close to Leica BLK360, Ouster), 900 nm (close to ibeo, Velodyne, Infineon Prototype), and 1560 nm (close to Leica RTC360/P30/P40/P50) were calculated for each spectrum. The resulting histograms are shown in Figure 13 and the mean and standard deviation of the difference in hemispherical reflectance for each material subclass are listed in Table 2.

The corresponding mean values for the wavelengths 840 nm and 900 nm are 0% with standard deviations of ±2% and ±1% respectively. The analysis of the wavelengths 660 nm and 1560 nm provide a larger difference.

The ECOSTRESS library samples represent hemispherical reflectance data given in %. To provide an incidence angle depending reflectance function $R_{\lambda }\left(\theta \right)$ for comparison with the TOF camera measurements, that fits to the above described lidar use case, in which both incident and reflected radiance have the same incidence angle θ, the following assumptions were applied:The reflectance values of the ECOSTRESS library corresponds to the above described reflectance value $R_{\lambda }\left(\theta =0^{\circ }\right)$ at incidence angle $\theta =0^{\circ }$.The reflectance function $R_{\lambda }\left(\theta \right)$ decreases with increasing incidence angle θ according to cos(θ). This takes into account that the illumination of a lidar decreases with increasing incidence angle according to cos(θ).

The incidence angle depending reflectance function $R_{\lambda }\left(\theta \right)$ based on the ECOSTRESS spectral library can therefore be derived by the following equation:(10)$$R_{\lambda }\left(\theta \right)=R_{\lambda }\left(\theta =0^{\circ }\right)cos\left(\theta \right).$$

Figure 11 shows the comparison TOF camera measurements and derived ECOSTRESS library values (based on Equation (Equation 10)) for all material subclasses that could be related to one or more spectra of the ECOSTRESS library.

## 8. Lidar Parametrisation

Section 3 introduces the reflectance limit function $R_{L}\left(r\right)$, that is required to apply the presented lidar modelling approach, and proposes a corresponding measurement concept. The parametrisation of the reflectance limit function $R_{L}\left(r\right)$ can be done either by conducting a measurement campaign or by using data that are provided in the specification sheet of some lidar types. The following Section further elaborates on the parametrisation procedure for the lidar modelling approach by presenting a measurement campaign to derive the reflectance limit function $R_{L}\left(r\right)$ for a new lidar prototype from Infineon and by presenting relevant data from lidar specification sheets of common lidar types.

### 8.1. Infineon Lidar Prototype Measurements

Lidar measurements were conducted according to the measurement concept described in Section 3 using a new Infineon lidar prototype (a successor of the lidar that was presented by Druml et al. [5]). The measurements were carried out in the Carissma indoor test facility in Ingolstadt. The Carissma facility has been chosen since it provides constant ambient light conditions and a large area of 100 m × 18 m for testing.

The working principle of the Infineon lidar prototype is depicted in Figure 14. The Infineon lidar uses a 1D MEMS laser scanning architecture. A laser bar with eight parallel edge emitting lasers projects a single solid horizontal line. A 1D scanning MEMS mirror is used to sweep the horizontal illumination line in a vertical direction. For each angular increment of the MEMS mirror a light pulse is sent out, covering a rectangular area. Objects located in the scene reflect a part of the light back towards a receiver, which collects the light on a 1D detector array. Each detector element corresponds to a specific row in the image taken of the scene, whereas each angle increment of the 1D MEMS mirror corresponds to a specific column. The receiver module detects the reflected laser pulse, amplifies the signal with transimpedance amplifiers (TIA) and converts it into a 1bit digital signal. The receiver module includes two arrays with 32 avalanche photodiodes (APD), four IFX 16 channels RX chip (containing TIA) and an IFX custom lenses.

The photons emitted by the transmitter are reflected by the objects in the scene, and a small portion returns back at the receiver. For each of the pixels in the 1D detector array, the time of flight is measured. Entire digital signal processing and components synchronization is handled by an FPGA. Both analog and digital data is collected, as well as the post-processed point-cloud lidar output. Analog (APD/TIA) and digital (Low Voltage Differential Signaling (LVDS)) data are used as input for a Matlab-framework to perform cross-check between simulation and real data, as well as testing different configuration and data processing parameters.

The setup for the lidar measurements is depicted in Figure 15 and included the following steps: three calibrated targets from SphereOptics [27] with defined reflectance values $R_{\lambda }\left(L\right)$ at 10%, 50%, and 95% and a size of 50×50 cm were mounted on a Tripod covered with absorbing tape. Each target was moved from five meters to the maximum measuring distance in five meter steps. At each position the following steps were repeated:Check the distance with external laser distance measurement tool (ground-truth).Align lidar vertically and horizontally using an IR camera to check where the laser is shooting and ensure that at least one APD channel is fully covered by the target’s reflection.Record analog (APD/TIA) and digital (LVDS) data with the lidar.Compare the lidar measurement of the relevant APD channel with the ground-truth data to evaluate whether the lidar detected the target or recorded a false alarm.

To compare the lidar measurements with the ground-truth data, a probability of false alarm, a probability of missed detection and a ranging accuracy value (standard deviation of the distance calculated from the correctly recognised targets) are calculated for each measurement. The parametrisation of the presented lidar modelling approach requires only a distinction of whether a false alarm is recognised or not, that is, whether the probability of false alarm is 0 or 1. If the measured lidar distance is equal to the measured ground truth including ambiguity-offset scaled with a factor based on the ground truth distance and maximum measured distance, no false alarm is recognised, hence the probability of false alarm is equal to 0. If the measured lidar distance is much smaller or larger than the measured ground truth including ambiguity-offset, a false alarm is recognised, hence the probability of false alarm is equal to 1. The reflectance limit $R_{L}$ for the respective Lambertian target is the last measurement that is not counted as false alarm, that is, it is still close enough to the ground-truth distance taking the ambiguity-offset into account.

### 8.2. Lidar Specification Sheets

The specification sheets of several common automotive lidar types include maximum range information depending on target reflectance. This data can directly be used to parametrise the lidar capabilities in the presented modelling approach. Figure 16 illustrates reflectance limit functions $R_{L}\left(r\right)$ derived from specification sheets and the suggested method from Section 3 for the following lidar types—Ouster OS-0 [23], Ouster OS-1 [24], Ouster OS-2 [25], Velodyne Alpha Prime [26], Ibeo Lux 4L/8L/HD [20], Leica P30/P40 [21], Leica P50 [22].

To evaluate the specification data provided by lidar vendors, measurement data were collected using an Ouster OS-1-64 [24] and three Lambertian targets from SphereOptics [27]. The Ouster OS-1-64 has a vertical resolution of 64 pixels and is able to record intensity values in addition to distance information for each point. The Lambertian targets have defined reflectance values $R_{\lambda }\left(L\right)$ of 10%, 50%, and 95% and a size of 50×50 cm. The following procedure was conducted three times, that is, for each Lambertian target:A person holding the Lambertian target with its surface normal pointing towards the Ouster OS-1-64, moves straight towards the Ouster OS-1-64, while the Ouster OS-1-64 is recording both distance and intensity values (Figure 17).The measurement points that are associated with the Lambertian target are separated from the remaining points by a combination of automatic and manual data processing. This requires at least four measurement points to locate the Lambertian target in the point cloud.

Figure 18 shows the intensity values *I* of the points associated with the three Lambertian targets plotted against distance. Measurement points above 37 m could not be associated to the Lambertian targets anymore. This makes sense, considering the target size of 50 × 50 cm, the beam divergence of the Ouster OS-1-64 and the requirements of the applied data processing chain for the detection of the Lambertian target in the point cloud. At range *r* = 0 m, the diameter of the laser beam is defined in the specification sheet with 9.5 mm and the beam divergence of the Ouster OS-1-64 is defined with $0.18^{\circ }$, which corresponds to 31.42 cm at 100 m. Hence, to derive the maximum detection range $d_{x}$ of the Ouster OS-1-64 for targets with reflectance values $R_{\lambda }\left(L\right)$ of 50%, and 95%, the size of the Lambertian target need to be at least 1 m × 1 m.

Figure 18 shows that the Lambertian targets with reflectance values $R_{\lambda }\left(L\right)$ of 50%, and 95%, can still be detected at 37 m distance, while the Lambertian target with reflectance value $R_{\lambda }\left(L\right)=10\%$ could only be detected until 30 m distance. Hence the measured maximum detection range $d_{x}$ of the Ouster OS-1 for a 10% reflectance target is 30 m. According to the specification sheet, the maximum detection range $d_{x}$ of the Ouster OS-1 for a 10% reflectance target should be between 40 m at >90% detection probability and 60 m at >50% detection probability.

## 9. Conclusions and Outlook

A new lidar modelling approach is presented that shall enable a more realistic representation of lidar sensors in virtual test environments by taking material properties (i.e., incidence angle dependent reflectance) and corresponding lidar capabilities (i.e., capability to detect a material with a certain reflectance up to a certain range) into account. The approach is suitable for both object and raw data based lidar models.

A material classification that includes the most relevant materials for lidar simulations in the automotive context is introduced. This allows to establish a reusable material data set that can easily be implemented into a virtual test environment, that typically provides a similar object and material distinction.

To collect angle dependent material reflectance measurements, a new measurement device based on a TOF camera is introduced and calibrated using Lambertian targets. The intensity measurements of the Lambertian targets decrease with increasing incidence angle θ according to cos(θ) and the intensity values increase linearly with increasing target reflectance. This shows that the measurement device behaves as expected and gives confidence in the TOF camera measurements.

Data from the NASA ECOSTRESS library is used to further evaluate the measurements of the new device.

Considering the spectra of almost 500 relevant materials reveals that there is only little difference between hemispherical reflectance values at 940 nm and hemispherical reflectance values at 900 nm and 840 nm. Hence, the TOF camera measurements at 945 nm may be used to represent the material behaviour at wavelengths used by several common lidar vendors that is, ibeo, Ouster, and Velodyne.

Comparing the hemispherical reflectance values at 940 nm to the hemispherical reflectance values at 660 nm and 1560 nm yields large differences (up to 60%) for certain materials. In particular, photosynthetic and non-photosynthetic vegetation show a strong variation in reflectance depending on wavelength. The strong change in reflectance between 940 nm and 660 nm in the vegetation data is linked to the known phenomenon ’red edge’. The chlorophyll in vegetation absorbs most of the visible light but becomes almost transparent at wavelengths greater than 700 nm. The cellular structure of the vegetation then causes high reflectance values because each cell acts like a corner reflector [34]. This behaviour is also well observable in Figure 12. Hence, the TOF camera measurements at 945 nm cannot be used to represent the reflectance behaviour at wavelengths around 660 nm and 1560 nm and are therefore not well suited for for example, Leica lidar products such as RTC360/P30/P40/P50. To cover these wavelength, the measurement device would need to be adopted and equipped with a sensor that allows to record intensity values at wavelengths closer to 660 nm and 1560 nm.

Based on the spectra from the NASA ECOSTRESS library mean, minimum and maximum reflectance values are calculated for several material subclasses. This data is then compared to measurements collected with the new measurement device. The comparison allows three main conclusions:Most TOF camera measurements are in the same order of magnitude as the NASA ECOSTRESS data. Hence, in most cases a single TOF measurement can be used to capture the correct order of magnitude of a certain material subclass, that is, around ±10% reflectance. However, as shown in Figure 16, a reflectance variation of ±10%, can affect a change in maximum detection range of several 10 m and in a view cases even up to more than 100 m.The NASA ECOSTRESS data show different spread in reflectance for different material subclasses. Hence, to provide a good representation of the reflectance spread within a material subclass, single TOF measurements are not sufficient, but a larger data set is required.The TOF camera measurements reveal the importance of angle dependent measurements in particular for materials with specular reflection behaviour such as metal or glass. Assuming Lambertian behaviour based on hemispherical reflectance is not sufficient to describe the angle dependent reflectance of such materials.

The parametrisation of a specific lidar in the presented modelling approach is done by defining its maximum range depending on the reflectance of the illuminated material. Several lidar vendors provide maximum range information depending on target reflectance in the specification sheet. Figure 16 provides an overview of the specification data provided by common automotive lidar vendors. An ADAS/AD function based on lidar environment perception can only rely on the lidar up to this specification limits, since the lidar vendor does not provide any guarantee for measurements collected outside the specification range. However, as shown in Figure 16, the information currently provided by common automotive lidar vendors typically consists of only one or two data points. The few values presented in the specification sheets are probably underestimating the real performance of a lidar under favourable conditions and overestimating the performance under unfavourable conditions. Information on the variation of lidar performance are required to provide a more profound basis for developing ADAS/AD functions based on lidar perception. In conclusion, the spare information currently provided by automotive lidar vendors (Figure 16) allow to parametrise the lidar within the legal performance limits on which an ADAS/AD function can rely on, but might not be sufficient for a good parametrisation of the real lidar performance.

Unlike automotive lidar vendors, manufacturers of high-end terrestrial laser scanners, such as RIEGL, typically provide very detailed data on the maximum measurement range of their products depending on target reflectance and external conditions, such as visibility. Figure 19 illustrates the maximum measurement range specification of the RIEGL VZ-6000 3D ultra long range terrestrial laser scanner [35] depending on target reflectance in %, scanning speed in kHz and visibility in km. A similar precise specification from automotive lidar vendors would be very helpful to both better define the performance limits on which the ADAS/AD function can rely on and for an easy and good parametrisation of the real lidar performance in the presented lidar modelling approach.

To conduct a more profound and realistic parametrisation for current automotive lidar types, a similar measurement campaign, as presented above with an Infineon lidar prototype, is required. The data that was collected in the presented measurement campaign with the Infineon lidar prototype are owned by Infineon AG and due to the prototype status of the used lidar device, the authors do not have the permission to publish the measurements as part of this publication. Nevertheless, the detailed description of the measurement procedure shall give a good insight on how the lidar modelling approach can be parametrised in a measurement campaign.

Environmental conditions are not yet taken into account in the presented modelling approach. Nevertheless, they can have a strong impact on both material properties (e.g., snow cover or wet surfaces can change the reflectance properties of a material significantly) and lidar capabilities (e.g., heavy rain can reduce the detection range significantly). Hence, a more realistic simulation of adverse environmental conditions can be achieved by adopting the material reflectance function $R_{\lambda }\left(\theta \right)$ of the illuminated surface and the reflectance limit functions $R_{L}\left(r\right)$ of the simulated lidar accordingly. In terms of material properties, a feasible option is to adapt the material classification by introducing an additional category ’attribute’ to each subclass, that states whether the surface is dry, wet or covered with snow, ice or dirt. Investigating the effects of adverse environmental conditions on lidar sensors for a better understanding of the limitations of this sensor type and for developing more realistic sensor models are currently ongoing research activities at the VIRTUAL VEHICLE research center.

For the sake of simplicity, the current lidar modelling approach suggests a discrete transition between detected and undetected region. However, various small factors, for example, thermal noise in the receiver, add uncertainty to the detection probability. To include this uncertainty, a probability distribution between detected and undetected region could be used instead of a discrete transition. This would require a more elaborate lidar measurement campaign that allows to build the probability distribution.

In addition to range information, several new lidar types, for example, Ouster OS [23,24,25], provide intensity information for each received point. As shown in Figure 20, the received intensity values can strongly vary depending on the reflectance of the illuminated material. New ADAS/AD functions might relay on this additional intensity information for object recognition, for example, detection of lanes or road markings. The presented lidar modelling approach can be extended to provide intensity values *I*, since the intensity values are also a function of material reflectance $R_{\lambda }\left(\theta \right)$, range *r* and lidar capabilities:(11)$$I=f(r,R_{\lambda }\left(\theta \right)).$$

Including intensity values into the presented modelling approach requires a lidar measurement campaign with defined test targets, that includes the collection of intensity values and allows to relate the intensity values to range and reflectance.

## Figures and Tables

**Figure 1 sensors-20-03309-f001:**
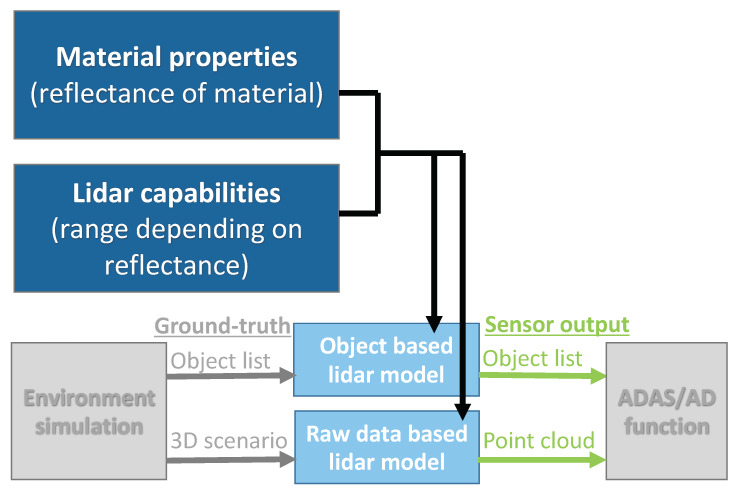
Schematic illustration of data input and output for lidar sensor models on object and raw data level. This publication proposes a new approach to include material reflectance and corresponding lidar capabilities into both object and raw data based lidar models.

**Figure 2 sensors-20-03309-f002:**
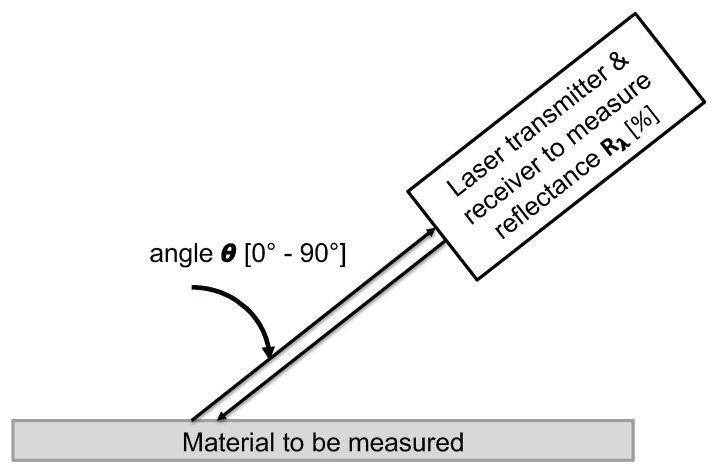
Measurement concept to derive the angle dependent reflectance $R_{\lambda }$ of a material depending on incidence angle θ.

**Figure 3 sensors-20-03309-f003:**
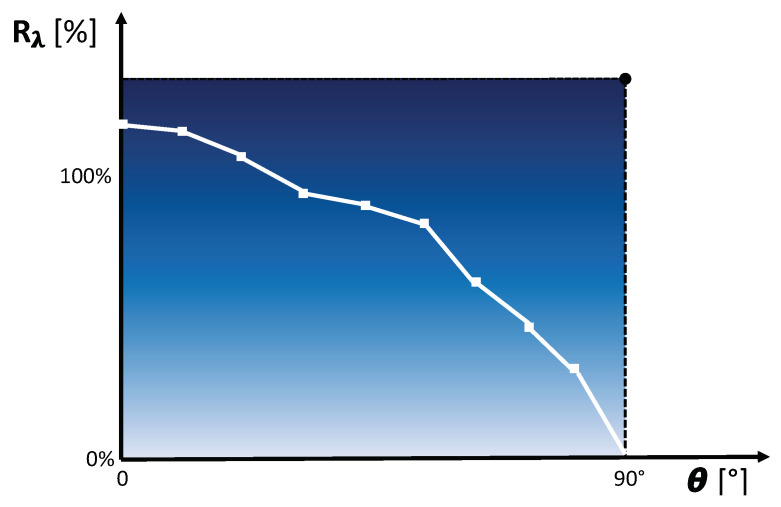
Exemplary reflectance $R_{\lambda }$ curve depending on the incidence angle θ.

**Figure 4 sensors-20-03309-f004:**
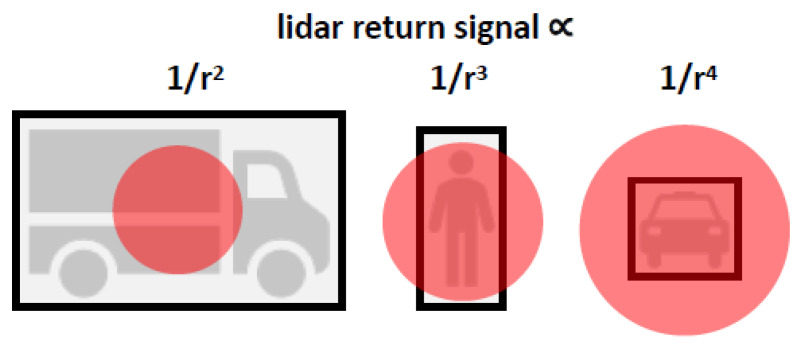
Schematic illustration how the relative size of the laser beam (illustrated by red circle) compared to the size of the illuminated surface (illustrated by black rectangles) affects the attenuation of the lidar return signal depending on range *r*.

**Figure 5 sensors-20-03309-f005:**
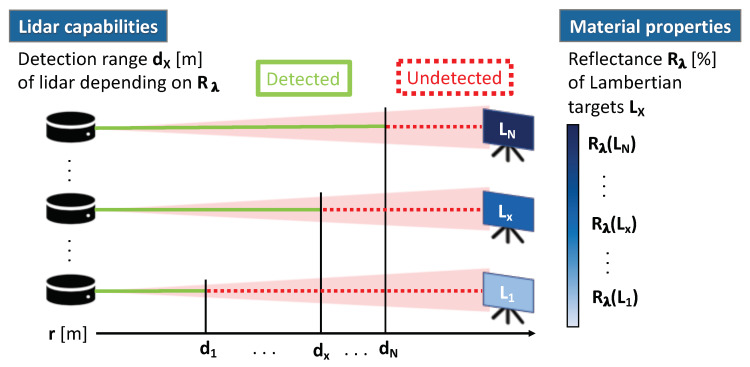
Measurement concept to derive the reflectance limits $R_{L}$ of a lidar: Lambertian targets $L_{X}$ with defined reflectance values $R_{\lambda }$ are placed at incidence angle $\theta =0^{\circ }$ at different distances to estimate the maximum detection range $d_{x}$ for each target.

**Figure 6 sensors-20-03309-f006:**
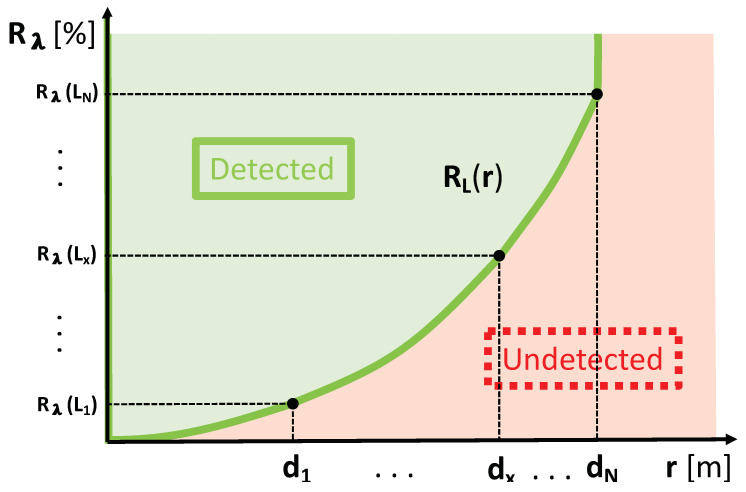
Exemplary reflectance limits function $R_{L}\left(r\right)$ depending on range *r*. Depending on the target reflectance *R* and the distance between target and lidar, i.e., range *r*, the target is either detected (green area) or undetected (red area).

**Figure 7 sensors-20-03309-f007:**
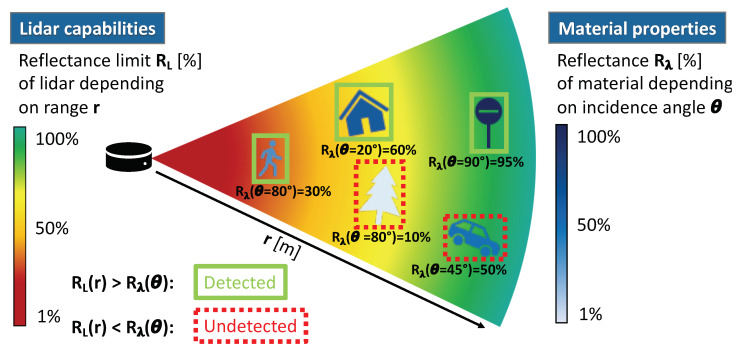
Lidar modelling approach taking material properties and lidar capabilities into account. Depending on whether the range *r* dependent reflectance limit $R_{L}$ of the lidar is above or below the incidence angle θ dependent reflectance $R_{\lambda }\left(\theta \right)$, the surface (raw data based lidar model) or object (object based lidar model) is detected or remains undetected.

**Figure 8 sensors-20-03309-f008:**
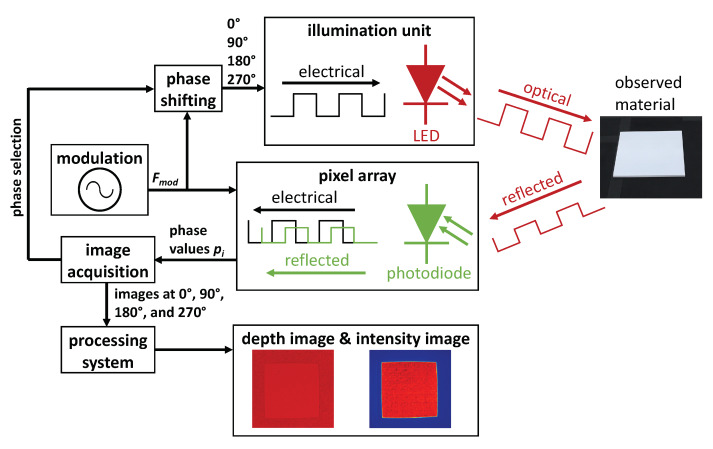
Measurement principle of the time-of-flight (TOF) camera including all processing units. The illustrated example shows a Lambertian target placed on a non-reflecting surface as illustrated in Figure 9 and the corresponding depth and intensity image taken with the TOF camera at incidence angle $\theta =0^{\circ }$.

**Figure 9 sensors-20-03309-f009:**
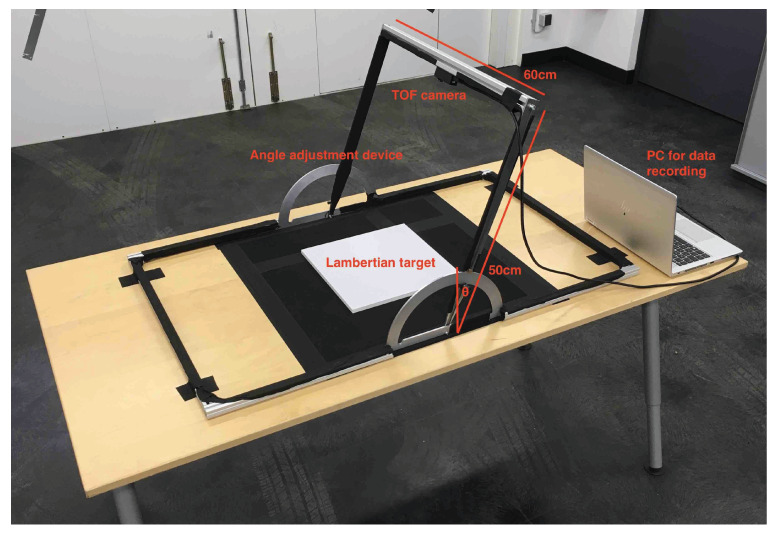
Reflectance measurement setup including the time-of-flight (TOF) camera, the angle adjustment device for setting the incidence angle θ and a PC for data recording. Lambertian targets, placed on a non-reflecting surface, were used for calibration.

**Figure 10 sensors-20-03309-f010:**
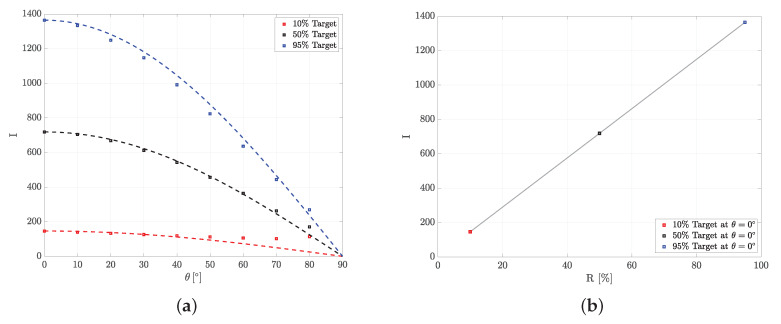
TOF camera measurements of Lambertian targets with defined reflectance values at 10%, 50%, and 95% [27]: (**a**) intensity *I* depending on incidence angle θ, the points represent the measurements and the dotted line the theoretical curve $I\left(\theta =0^{\circ }\right)\times cos\left(\theta \right)$, and (**b**) calibration curve linking intensity *I* with reflectance *R* (%) of Lambertian targets at incidence angle $\theta =0^{\circ }$.

**Figure 11 sensors-20-03309-f011:**
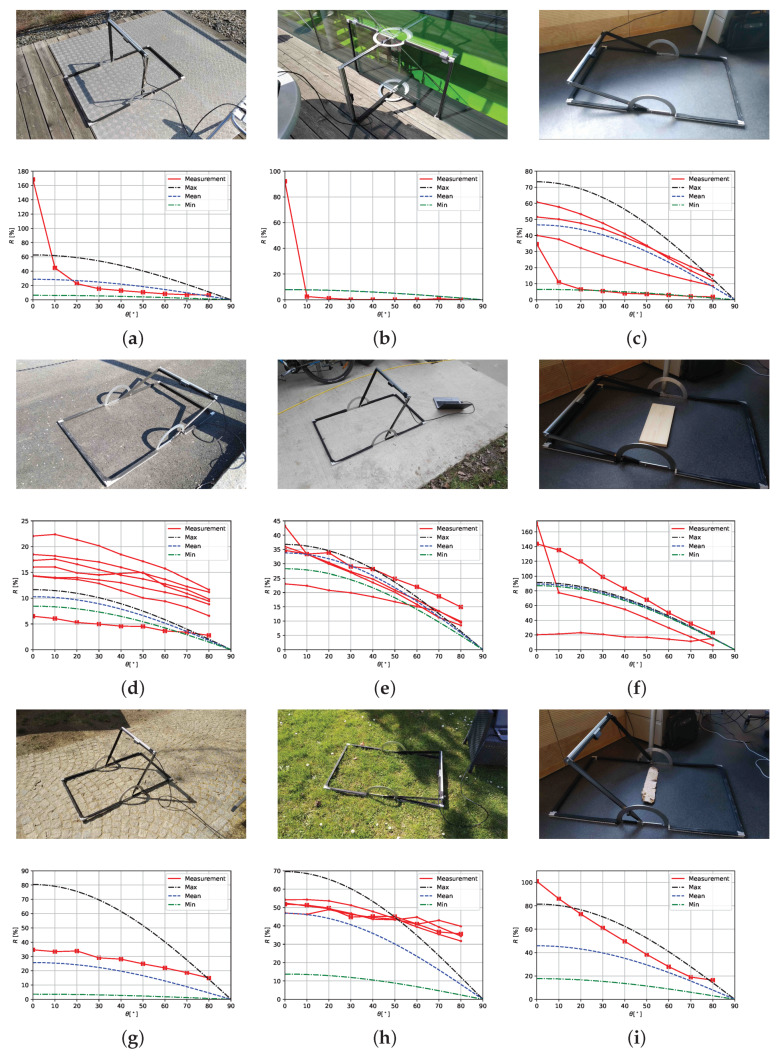
Images of observed materials together with the measurement setup and corresponding reflectance values *R* (%) depending on incidence angle θ. The depicted reflectance values are measured by the TOF camera (red, the measurements marked with rectangles are related to the images above) and derived from the NASA ECOSTRESS library (mean, minimum, and maximum in blue, green, and black) assuming a reflectance decrease of cos(θ). Following material subclasses are depicted: (**a**) metal, (**b**) glass, (**c**) rubber, (**d**) asphalt, (**e**) concrete, (**f**) wood, (**g**) rock, (**h**) photosynthetic vegetation, and (**i**) non-photosynthetic vegetation. Note the different scales of the *Y*-axis.

**Figure 12 sensors-20-03309-f012:**
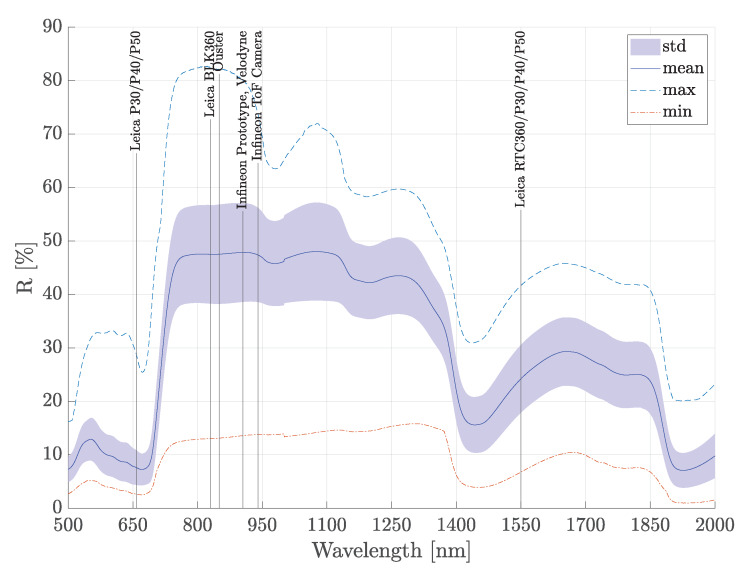
Minimum, maximum and mean hemispherical spectral reflectance values incl. standard deviation based on the ECOSTRESS spectral library for the material subclass photosynthetic vegetation including the wavelengths of common lidar types and the used TOF camera.

**Figure 13 sensors-20-03309-f013:**
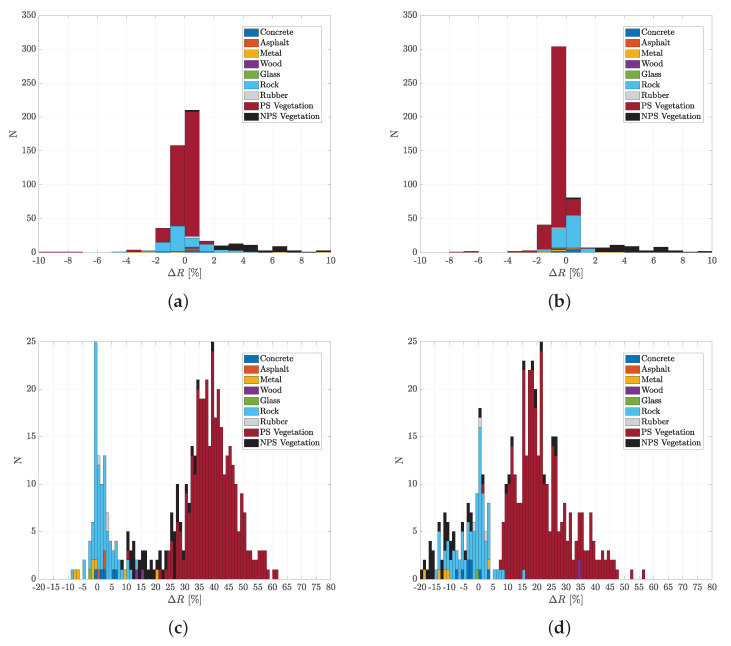
Difference of hemispherical reflection value at 940 nm (close to TOF camera) and (**a**) 840 nm (close to Leica BLK360, Ouster), (**b**) 900 nm (close to ibeo, Velodyne, Infineon Prototype), (**c**) 660 nm (close to Leica P30/P40/P50), and (**d**) 1560 nm (close to Leica RTC360/P30/P40/P50) for each material subclass (PS … photosynthetic; NPS … non photosynthetic) based on the ECOSTESS libray. Note the different scales of the axis.

**Figure 14 sensors-20-03309-f014:**
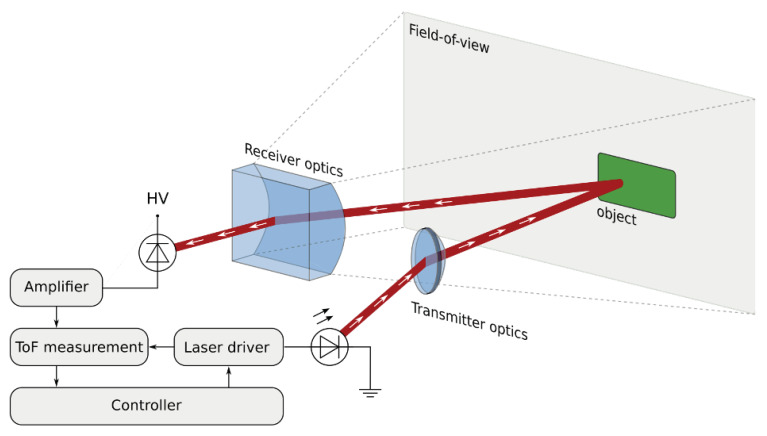
Architecture overview and working principle of the Infineon lidar prototype including all relevant components.

**Figure 15 sensors-20-03309-f015:**
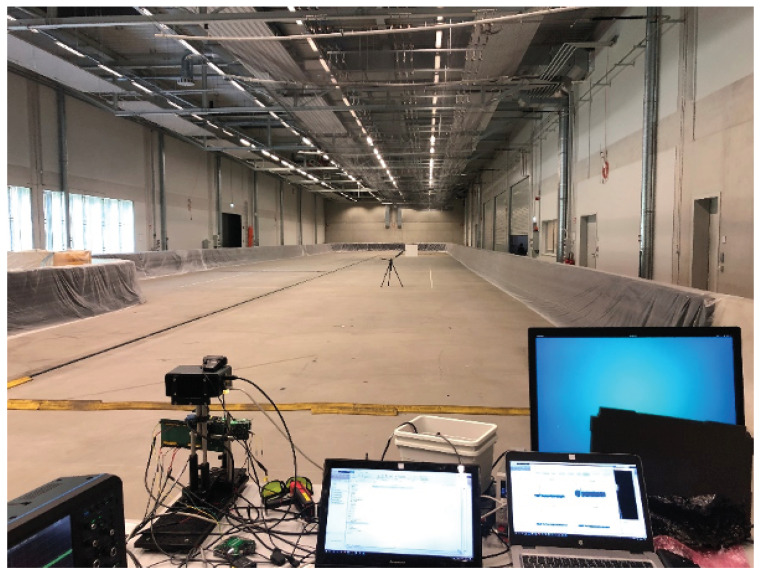
Lidar measurement setup in the indoor test-facility Carissma. The Infineon lidar prototype is pointed towards a Lambertian target that is mounted on a tripod and placed at different distances.

**Figure 16 sensors-20-03309-f016:**
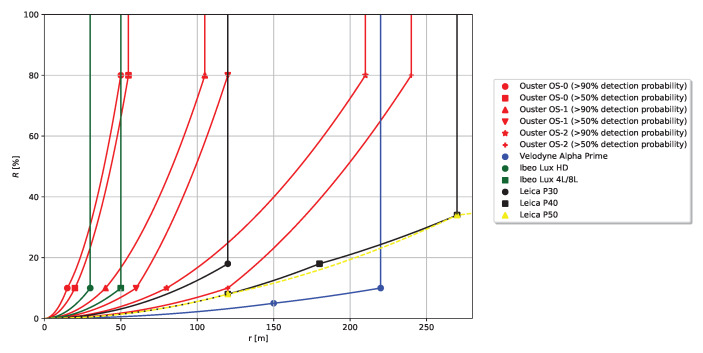
Reflectance limit functions $R_{L}\left(r\right)$ for common lidar types. The points depict the data values from the respective lidar specification sheet. Note that two points are missing for presentation purposes: Leica P50 has two additional data points (*r*, *R*) at (570 m, 60%) and (1 km, 80%). The solid and dotted lines connect the origin with the data points and illustrate the cut-off after the last data point according to the suggested method from Section 3.

**Figure 17 sensors-20-03309-f017:**
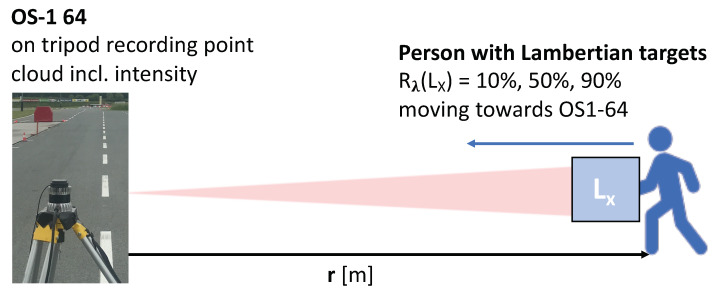
Schematic illustration of measurements setup including the Ouster OS-1-64 [24] and three Lambertian targets from SphereOptics [27] with defined reflectance values $R_{\lambda }\left(L\right)$ at 10%, 50%, and 95%.

**Figure 18 sensors-20-03309-f018:**
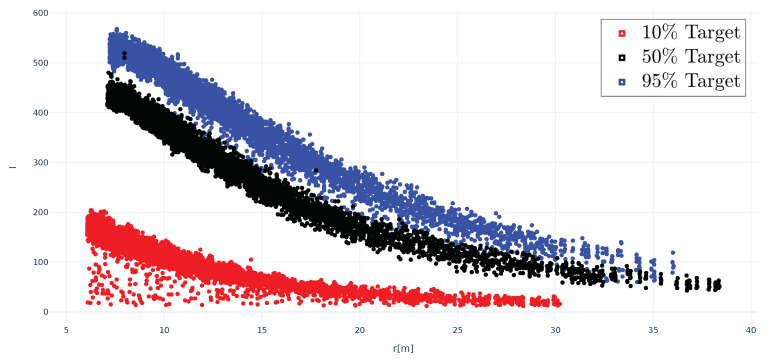
Ouster OS-1-64 intensity values *I* of the points associated with the three Lambertian targets against distance *r* (m).

**Figure 19 sensors-20-03309-f019:**
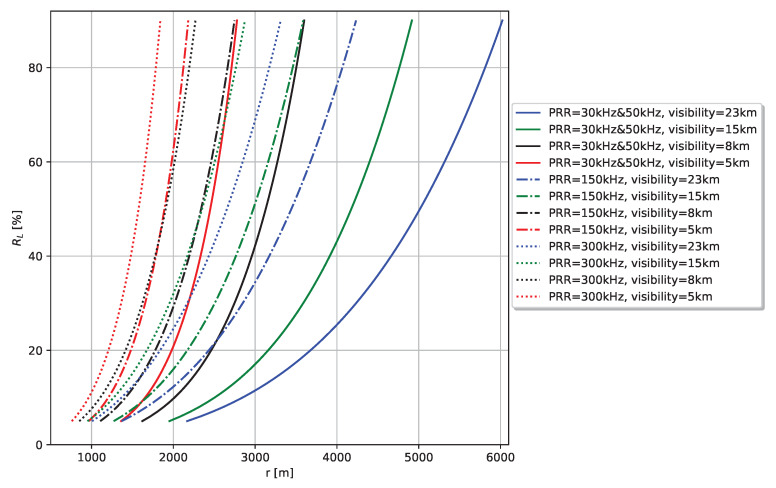
Maximum measurement range specification of RIEGL VZ-6000 3D Ultra Long Range Terrestrial Laser Scanner depending on pulse repetition rate (PRR) and visibility (23 km … standard clear atmosphere, 15 km—clear atmosphere, 8 km—light haze, 5 km—medium haze). Data was provided by RIEGL Laser Measurement Systems GmbH.

**Figure 20 sensors-20-03309-f020:**
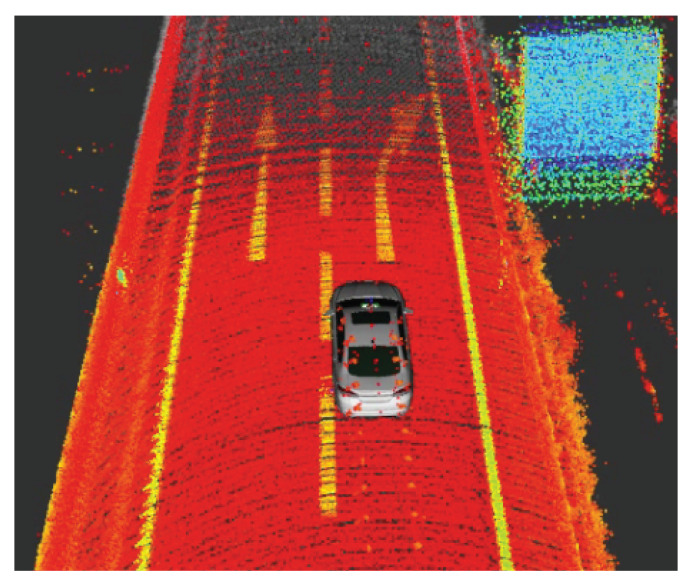
Cumulative lidar point cloud collected with an Ouster OS-1 lidar with 64 layers [24] on a highway junction near Graz, Austria. The colour scale depicts low (red), medium (yellow), high (green) and very high (blue) received intensity.

**Table 1 sensors-20-03309-t001:** Material list and classification of materials into type, class, and subclass. The numbers in brackets refer to the amount of spectra from the NASA ECOSTRESS library that were used for evaluation of the time of flight (TOF) camera measurements (Section 7).

Type	Class	Subclass
dynamic	vehicle	paint
		metal (7)
		glass (1)
		plastic
		license plate
		reflector
		rubber (3)
	pedestrian	skin
		clothing
	animal	fur
static	road	asphalt (3)
		road marking
		offroad track
	traffic sign	reflecting
		non-reflecting
	construction	concrete (5)
		glass (see above)
		metal (see above)
		wood (2)
	nature	rock (84)
		soil
		photosynthetic vegetation (336)
		non-photosynthetic vegetation (47)

**Table 2 sensors-20-03309-t002:** Mean and standard deviation of the difference in hemispherical reflectance depending on wavelength for each material subclass based on 488 spectra from the ECOSTRESS library. Values are given in % reflectance. NB: PS … photosynthetic; NPS … non photosynthetic; hemispherical reflectance of asphalt was not available at 1560 nm.

Material Subclass	$R_{940\;\mathrm{nm}}-R_{660\;\mathrm{nm}}$	$R_{940\;\mathrm{nm}}-R_{840\;\mathrm{nm}}$	$R_{940\;\mathrm{nm}}-R_{900\;\mathrm{nm}}$	$R_{940\;\mathrm{nm}}-R_{1560\;\mathrm{nm}}$
concrete	2.80±2.90	0.51±1.02	0.27±0.35	−2.90±3.35
asphalt	1.80±1.15	0.52±0.31	0.14±0.23	-
metal	1.66±9.93	1.33±4.97	0.48±1.95	−14.22±9.76
wood	14.51±0.89	0.10±0.01	−0.45±0.06	34.27±0.38
glass	−2.51±0	−0.73±0	−0.54±0	−0.85±0
rock	1.59±3.58	0.01±1.40	0.14±0.56	−1.95±5.60
rubber	2.59±2.12	0.21±0.27	−0.11±0.17	0.29±2.02
PS vegetation	39.86±7.33	−0.10±1.02	−0.44±0.81	22.37±9.41
NPS vegetation	21.96±7.65	5.15±2.54	5.14±2.53	−4.17±15.16

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
