# Peer review of "Automotive Lidar Modelling Approach Based on Material Properties and Lidar Capabilities"

_sensors, 2020, doi:10.3390/s20113309_

Round 1

Reviewer 1 Report

In this paper, Stefan Muckenhuber et al. developed empirical models for LiDAR, to account for the correlation between sensor detectability,  material reflectance, and incidence angle. The authors provided a good coverage of the background, and most of the existing theories which the new model is based on. The models being developed appear to be mostly sound and consistent with the experimental data, which however should have been more sufficient than what it is.

Last but not least, a few points to be improved are listed as follows:

  1. (Line 118-119). "... is then called reflectance limit RL". The notation RL is misleading. It can be found confusing by readers and misinterpreted as "R times L". Subscription might be clearer.
  2. (Line 122-123). "the maximum detection range dx". Some backgrounds and definitions about maximum detection range should be provided. The reviewer himself also found several different definitions coexist in the current literature, from different perspectives of optics, system engineering, as well as perception. It appears to me that authors use Equation (4) on Page 7 as the definition of maximum detection range. However, this definition is only based on a reflectance threshold, and oversimplifies the detectability to 0 or 1, in contrast to the reality it is more close to a probability over a wide range. At least some discussion should be provided here.
  3. Data points in Figure 15 are too sparse, and is directly taken from manufacuturer's data sheets, which can be not scientific rigorous. More experimental data is expected here.

Thank you!

Reviewer

Author Response

Dear Reviewer # 1,

Thank you very much for helping us improving our paper!

Please find here attached the answers to your comments and the corresponding changes in manuscript.

Thanks again for your comments. We are looking forward to your reply!

Best regards,

Stefan Muckenhuber

Reviewer 2 Report

The scientific approach proposed is coherent but remains unsuccessful because the work carried out should have been improved. The reader could be disappointed by lack of interpretation and the results are sometimes underused or analyzed. The paper is however well referenced and well written.

We first question certain notions defined by the authors to characterize the optical response of a surface to a LIDAR and define classification.

The authors assess the detection range (i.e. the minimum signal) on Lambertian surfaces of variable reflectivity value. With the apparent angular reflectivity defined beforehand, they can estimate in the sensor model whether a material X will be detected or not by the Lidar for a given distance. This approach remains theoretical and unfortunately is not tested with real data in §8. Furthermore, as we will discuss next, the linear approximation to define detection or range limits with only 3 measurement points requires justification.

The choice of materials to build the database is consistent. The idea of supplementing with an ECOSTRESS database is a priori a good idea but questions.We also have comments on the methodology used and the results obtained.

The authors present the well-known principle of an AMCW TOF camera. This part could perhaps have been lightened using references. The approach of using this camera to make relative measurements of apparent reflectivity (at the wavelength of the TOF camera, i.e. 940nm) is interesting. The calibration procedure seems consistent with satisfactory results. The applicability of this type of measurement to the parametrization of all types of LIDAR (regardless of spectral variability) remains debatable and should at least have been discussed.

Using a database to spectrally extend the apparent reflectance measurements made with the TOF camera was a good idea. But we do not understand why the authors are focusing here only on materials from ECOSTRESS for the "spectral" aspect? Paints (car, road, buildings…) or traffic signs, present also a large spectral and angular variation…. For example, using the "roofing metal" ASTER data as representative of an ADAS scene is a bit surprising.

The authors have measured the reflectivity of materials close to the ECOSTRESS classes and compare their results with reflectivity estimation (average, min / max) assuming Lambertian behavior. We can first note that Lambertian tendency of most of measured materials is not established. Contrary to the author’s thinking, using the ECOSTRESS base seems to be for us not sufficient validate the applicability of a TOF camera measurement to characterize surfaces for LIDAR parameterization and classification of surfaces encountered. We will discuss this point in particular.

Chapter 8 could have been the most interesting part of the publication since its aims was to confront the method with a real system. It first presents the system (a lidar prototype) that will be used for the tests and applies the procedure to determine RL for this lidar. Nevertheless, it seems unfortunate that the result obtained is not presented, commented or exploited in this paper. We do not understand why with an RL measured in the "laboratory", why targets characterized with the TOF camera (as the spectral difference of laser emission is small) were not tested to evaluate the consistency of the proposed approach.

For all this reasons, we therefore propose some additions and modifications in order to improve the comprehension of the readers and / or to correct  lack and ambiguity.

In § 1

COMMENT 1: The range performance of a lidar on a surface cannot only be related to an angular reflectance. The effects of the atmosphere (in particular in degraded visibility conditions) have an impact on performance (Rasshofer et al., Advances in Radio Science, vol. 9, pp. 49–60, 2011) as well as environmental effects (e.g. solar flux received by the LIDAR, modification of the backscattering of a wet surface, etc.). A sensor Lidar model coupled with a complete ADAS simulation tools have to integrate under penalty of overestimating the Lidar performance. The authors do not intend to focus on this point in the work presented and they position these limits in the conclusion. We suggest to the authors to define the limits of their approach in the introduction and to indicate that these limits will be discussed in perspectives.

In § 2

COMMENT 2: the exact coefficient for characterizing the radiometric response of a surface to a monostatic lidar and taking it into account in a radiometric assessment (in terms of performance) is the spectral backscatter coefficient (or more generally the BRDF) and its unit is the sr-1. This is the relationship between a spectral radiance (W / m3 sr) reflected in the direction of the sensor and an incident spectral laser irradiance (W / m3) (with the same direction in the case of a monostatic lidar). Unless justified, it is difficult to understand the use of a spectral reflectivity.

COMMENT 3: spectral reflectivity is mostly defined as a radiant power or irradiance ratio and very rarely as a radiance ratio. We do not understand the choice made by the authors, which is not representative of the measurement made by the TOF camera. This ratio is always less than or equal to 1.

COMMENT 4: the authors define the value of spectral reflectance in relative by the use of a material known to be Lambertian and quasi-invariant spectrally whose backscatter coefficient is written as Rl / π with Rl its spectral hemispheric reflectance. The apparent spectral reflectivity of a material is therefore defined and subsequently measured according to the following expression: TOF camera signal measured at the wavelength and at the incidence q on material X divided by the TOF signal at the wavelength measured on a Lambertian material in nadir incidence multiplied by the spectral reflectivity value (hemispherical) of the Lambertian material at the wavelength. With this definition, this apparent reflectance can indeed be greater than 1. With the notations and definitions they used, the authors bring some confusion and we suggest that they clarify this.

In § 4

COMMENT 5: The Authors state: “As stated in Section 1, current lidar models typically do not take material properties and corresponding lidar capabilities into account”. This statement is too exclusive and needs to be clarified. First of all, there are many lidar simulation models that take into account realistic 3D scenes from a physical point of view, i.e. given in optical properties: ex: DIRSIG in the USA. Few ADAS simulation tools taking into account reflectivity (even bad weather) were published (ex: Goodin et al., Electronics 2019, 8, 89) and commercial products seem to propose this capability ( ex: https://www.tesis.de/en/sensorsimulation/?r=1). We would appreciate it if the authors extended their analysis a little considering these elements.

COMMENT 6: A linear model is an acceptable interpolation assuming enough points to build it. Figure 5 is constructed with 3 reflectivities and in § 8, it seems that only few points are used. How can the authors justify the use of a linear interpolation to delimit the RL (r) detection curves?

COMMENT 7: From a radiometric point of view only, the LIDAR will detect if its signal exceeds a certain limit. The LIDAR signal varies as the apparent spectral reflectivity Rl (q) divided by the squared distance. Why the authors don’t use this property instead of “linear” interpolation.

In § 6

COMMENT 8: Can the authors specify the parameters of the TOF camera that will impact on the measurement made: divergence, field of view, measurement technique, etc. (cf. COMMENT 9 below).
COMMENT 9: Could the authors discuss the generalization of their approach to any lidar technique? Indeed, the TOF camera used illuminates on a large FOV with a divergent source which averages the backscatter effects both angularly and on a surface. What about the applicability of their results for a LIDAR having less divergent laser as on few laserscanners like IBEO / OUSTER systems? the authors could for example study the work of BASISTYY et al. (Vol. 57, No. 24/20 August 2018 / Applied Optics) where it can be observed on asphalt but also other urban materials an important effect of angular or spatial selectivity...

COMMENT 11: We also suggest some clarifications on the TOF measurement. It would be useful for the authors to specify how the value of the intensity I is obtained: is it the average value of the all image pixels or an average made on some chosen pixels (if so which ones?). Contrary to the Authors’ claims, the 10% material clearly does not have a Lambertian behavior. It should be clarified. It would have been useful to plot, in addition to the absolute intensity values I an evolution of normalized intensity with respect to the value at normal incidence to compare with a theoretical curve cos q. It is an indicator of the deviation of a Lambertian behavior but gives the instrument function assuming calibration with a true Lambertian material. The authors should discuss this particular point

In § 7

COMMENT 12: Have the authors considered other spectral databases? Why did they choose ECOSTRESS? The question is how to generalize the trends and conclusions when they have been based only on a part of the targeted ADAS classes. Can the authors comment on this point?

COMMENT 13: the measurement protocol described (xenon lamp, incidence 23 °, emergence 27 °) is only valid for a particular measurement for ECOSTRESS data and does not apply to all of the materials compared next. What is the usefulness here of this information which induces a bias of understanding?

COMMENT 14: What are the ECOSTRESS data compared spectrally in Figure 12? The figure 11 represents hemispherical directional spectral reflectance. Can the authors clarify things?

COMMENT 15: Figure 12 could be difficult to interpret because it compares classes with many samples (spectra) to others with few samples. Doesn’t it induce an analytical bias? Strictly speaking, should the spectral analysis have been done with relative abundances of these materials on typical ADAS scenes?

COMMENT 16: Trends observed on the vegetation are linked to the well-known phenomenon of the “red edge” where the water content affects the reflectance.

COMMENT 17: The authors claim to make a representative measure of reflectivity at an order of magnitude. Having a 10 factor on a reflectivity induces a factor of 3 on the detection range… and therefore strongly affects the values of RL. Can the authors discuss this point?

In § 8

COMMENT 18: The authors should describe how the probability of false alarm and the probability of missed detection are calculated?

COMMENT 19: Where are the RL (r) results of this new INFINEON prototype? They are not presented in Figure 15 ... Can the authors explain this point?

COMMENT 20: How do the authors suggest using Figure 15 that is neither commented on nor used afterwards?

Author Response

Dear Reviewer # 2,

Thank you very much for helping us improving our paper.

Please find here attached the answers to your comments and the corresponding changes in manuscript.

Thanks again for your comments. We are looking forward to your reply!

Best regards,

Stefan Muckenhuber

Round 2

Reviewer 2 Report

I thank the authors for taking my comments into account. The corrections made and the new results added greatly improved this article which is now fully acceptable for publication to the scientific community. The authors tried to make up for the lack of publishable measures on the prototype infineon with other data and the effort is appreciable. The idea of inciting the ADAS lidar community towards more effort to quantify lidar performances according to the backscattering power of surfaces and also in degraded conditions like for TLS / ALS (Terrestrial / aerial Laserscanner) is welcome and I fully support this approach.